# Effective population size for culturally evolving traits

**Dominik Deffner**[1,2,3]*, **Anne Kandler**[1], **Laurel Fogarty**[1]

**1** Department of Human Behavior, Ecology and Culture, Max Planck Institute for Evolutionary Anthropology, Leipzig, Germany, **2** Science of Intelligence Excellence Cluster, Technical University Berlin, Berlin, Germany, **3** Center for Adaptive Rationality, Max Planck Institute for Human Development, Berlin, Germany

* deffner@mpib-berlin.mpg.de

**Data Availability Statement:** This manuscript does not contain any empirical data. Simulation and plotting code necessary to reproduce all results and figures in the manuscript can be found on

## Abstract

Population size has long been considered an important driver of cultural diversity and complexity. Results from population genetics, however, demonstrate that in populations with complex demographic structure or mode of inheritance, it is not the census population size, $N$, but the effective size of a population, $N_e$, that determines important evolutionary parameters. Here, we examine the concept of effective population size for traits that evolve culturally, through processes of innovation and social learning. We use mathematical and computational modeling approaches to investigate how cultural $N_e$ and levels of diversity depend on (1) the way traits are learned, (2) population connectedness, and (3) social network structure. We show that one-to-many and frequency-dependent transmission can temporally or permanently lower effective population size compared to census numbers. We caution that migration and cultural exchange can have counter-intuitive effects on $N_e$. Network density in random networks leaves $N_e$ unchanged, scale-free networks tend to decrease and small-world networks tend to increase $N_e$ compared to census numbers. For one-to-many transmission and different network structures, larger effective sizes are closely associated with higher cultural diversity. For connectedness, however, even small amounts of migration and cultural exchange result in high diversity independently of $N_e$. Extending previous work, our results highlight the importance of carefully defining effective population size for cultural systems and show that inferring $N_e$ requires detailed knowledge about underlying cultural and demographic processes.

## Author summary

Human populations show immense cultural diversity and researchers have regarded population size as an important driver of cultural variation and complexity. Our approach is based on cultural evolutionary theory which applies ideas about evolution to understand how cultural traits change over time. We employ insights from population genetics about the "effective" size of a population (i.e. the size that matters for important evolutionary outcomes) to understand how and when larger populations can be expected to be more culturally diverse. Specifically, we provide a formal derivation for cultural effective

GitHub: https://github.com/DominikDeffner/
CulturalEffectivePopulationSize.

**Funding:** This research has been supported by the
Max Planck Society and by the Deutsche
Forschungsgemeinschaft (DFG, German Research
Foundation) under Germany's Excellence Strategy
– EXC 2002/1 "Science of Intelligence" – project
number 390523135. The funders had no role in
study design, data collection and analysis, decision
to publish, or preparation of the manuscript.

**Competing interests:** The authors have declared
that no competing interests exist.

population size and use mathematical and computational models to study how effective
size and cultural diversity depend on (1) the way culture is transmitted, (2) levels of migra-
tion and cultural exchange, as well as (3) social network structure. Our results highlight
the importance of effective sizes for cultural evolution and provide heuristics for empirical
researchers to decide when census numbers could be used as proxies for the theoretically
relevant effective numbers and when they should not.

## Introduction

Cultural evolutionary dynamics are governed by individual-level cognitive processes and
demographic properties of the population [1, 2]. Archaeologists and anthropologists have
been particularly interested in the ways population size might shape cultural processes (see [3,
4] for recent reviews). When researchers consider the impact of population size on cultural
evolution, they predominantly refer to the number of individuals in a population. This census
population size $N$ is readily observable in real-world situations and can be quantified by count-
ing how many people are present at a certain place and time. Results from population genetics,
however, have long demonstrated that in most real-world populations it is not this census size,
but the effective size, $N_e$, that is the correct measure to use when calculating important evolu-
tionary parameters such as genetic diversity and divergence times between populations [5].

### What is effective population size and why do we need it?

The effective population size is a theoretical construct that links complex populations to sim-
pler, idealised models. This way, the effective size makes it possible to directly compare any
number of complex populations—each with their own complicating factors—in a way that
would otherwise be impossible. A commonly used simplified model in population genetics is
the Wright-Fisher model [5–7], and much of what we understand about evolution comes from
our understanding of evolution in such idealised models. The effective population size is
defined in relation to this model as the size of an ideal Wright-Fisher population that experi-
ences genetic drift at the same rate as a particular study population (see below for details).

To understand what we gain from the effective size, even if we are not particularly inter-
ested in Wright-Fisher models, let us assume there are two populations, A and B, that produce
a particular cultural trait with infinitely many possible variants. We now want to know
whether population size affects the number of different variants in a population. Population A
has a larger census population size of 1000 individuals, population B has a smaller census size
of just 500. Based on some theoretical model of a cultural evolutionary process, we might be
led to expect population A to display a greater number of variants than population B at that
trait. Can we expect to find this demographic relationship in data on census population sizes
and cultural diversity indices from both populations? The answer is that—regardless of how
good the model is—the relationship is unlikely to be found unless the ways in which the trait
of interest is transmitted in these real populations are otherwise identical in evolutionarily
important ways. Two populations with the same census size might not display the same num-
ber of cultural variants at a trait if they are marked by different demographic histories or if the
way in which the trait is transmitted differs among the populations (see [8, 9]). In other words,
the number of variants displayed in the two populations of equivalent census size will differ if
the effective population size of the trait in question differs between the two populations. In
such cases, the populations are not directly comparable, except through their relation to a sim-
pler model—through their effective population sizes.

Imagine we now discover that, 10 generations ago, population A had a population bottleneck where its census size fell to only 10 individuals before recovering to its current value of 1000. Genetic evolution will be affected by this bottleneck for a number of generations (culture might recover from such events much faster than genetics [10]). Both populations are otherwise identical and conform to the assumptions of the Wright-Fisher model, which we detail below. Accordingly, the effective population size of the small, stable population B is 500, the same as its census size. The effective size of population A, however, is only around 92 (see S1 Text for calculation). We can now use results from population genetics to calculate how many cultural variants we expect to see in each population, given certain transmission mechanisms and innovation rates. For population B with $N_e$ = 500, the expected number of unique variants, with unbiased transmission and an innovation rate of $\mu$ = 0.1, is 223. For population A with $N_e$ ≈ 92, we expect to see on average 41 variants in a given generation (see S1 Text for full details). Thus, although a relationship exists between effective population size and cultural diversity, a straightforward relationship does not exist between census size and diversity. Employing census size or more informal definitions—rather than effective population size—to predict cultural diversity is bound to lead one astray in any empirical population that violates one or more of the conditions of the Wright-Fisher model. As real populations differ from one another and from the assumptions of an ideal Wright-Fisher model in numerous evolutionarily important ways, the ability to unify and compare them is invaluable. This is relevant for any question where we need to understand the effects of drift, innovation, migration (intergroup transmission), and selection in cultural processes across populations.

For many animal and plant species, researchers have investigated the relationship between observed census size $N$ and calculated (genetic) effective size $N_e$. In one large-scale study, ratios of $N_e$ to $N$ were found to vary between 0.19 in a species of pine tree to 3.69 in a species of mosquito [11]. This demonstrates that, across species, a large range of relationships between census and effective sizes are possible and $N_e$ can also exceed the census size $N$ [12]—a possibility we expand on below. Estimates of this ratio for humans suggest that the genetic effective size is considerably lower than our census size with an $N_e$ of around 10,000 compared to a census size of almost 8 billion [13], possibly reflecting population bottlenecks in the past. All of this strongly suggests that in order to understand how demography affects cultural evolution, and which empirical comparisons are meaningful, we need to gain a better understanding of the cultural equivalent of $N_e$.

## Cultural effective population size—History and outline

The importance of effective population size has been partially acknowledged in the cultural evolution literature and the concept is often invoked. Henrich and colleagues [14], for example, write: "The theory explicitly predicts that it is the size of the population that shares information—the effective cultural population size—that matters, and if there is extensive contact between local or linguistic groups, there is no reason to expect census population size to correspond to the theoretically relevant population". More recently, Derex and Mesoudi [3] also claim that effective population size "depends on both population size and interconnectedness". However, the definition of $N_e$ as "the population that shares information" is not always correct and corresponds more closely to the 'breeding population' rather than the effective size. We hope to show that it is not only the number of individuals sharing information, also called "cultural equivalent $N$" [1], or the "number of teachers" [15], but the exact details of how information is passed on between individuals that should be expected to influence cultural effective population size. Furthermore, it remains unclear whether and how different forms of population interconnectedness as well as social network characteristics might influence effective

population size for culture, though intuition suggests that this influence may be strong. In a first formal treatment for cultural evolution, Premo [8] used simulation models to investigate the relationship between census size, effective population size, and the rate of change in mean skill level under non-ideal conditions. In the context of models by Shennan [16] and Henrich [17], the results show that natural and cultural selection can weaken the relationship between census population size and the rate of change in mean skill level. Further, Rorabaugh [15] and Premo [9] studied the accumulated copying error model (ACE) and investigated the relationship between the effective size and the coefficient of variation of a continuous trait passed via different transmission pathways. Importantly, every cultural trait has its own effective number depending on how it is transmitted [8, 9]. This adds a further complication, because each trait needs to be studied on its own and data regarding how trait X is passed may not be pertinent to trait Y even if both traits are in the same population. This work did not formally derive an appropriate measure for cultural effective size and examined the influence of selection and transmission pathways within the scope of rather domain-specific models.

Here we aim to provide this formal derivation and systematically examine the concept of effective population size for cultural evolution. We first introduce drift and effective population size as employed in standard models of population genetics. After deriving appropriate formulations of $N_e$, we use mathematical models and simulation to investigate how cultural $N_e$ depends on (1) the way traits are learned, namely one-to-many and frequency-dependent (i.e conformist and anti-conformist) transmission, (2) population connectedness through either migration or cultural exchange, and (3) social network structure. In each case, we relate effective numbers to the emerging levels of cultural diversity. We conclude by discussing implications for the role of demography in cultural evolution and provide heuristics for empirical researchers to decide when census numbers might be used as proxies for the theoretically relevant effective numbers.

## Drift and effective population size in population genetics

We first provide a basic introduction to the Wright-Fisher population, genetic drift, and the concept of effective population size as developed for genetic evolution.

### The Wright-Fisher population and genetic drift

The classic Wright-Fisher population is a closed, randomly mating population of $N$ individuals without selection and mutation [5–7, 18]. For each discrete generation, new individuals are formed by random sampling, with replacement, of gametes produced by the previous generation. The number of offspring for a given member of the parental generation is a binomially distributed random variable with both mean and variance of approximately 1 (for haploid populations; see S2 Text for explanation). Genetic drift describes the random change in allele frequencies from generation to generation by the chance success of some alleles relative to others [19–21]. While per definition unpredictable in any particular instance, on average, drift causes populations to change in broadly systematic ways: drift reduces the number of alleles in a population, increases the differences among populations and results in higher variability of allele frequencies over time. Crucially, the magnitude of allele frequency changes due to genetic drift is inversely related to the size of the Wright-Fisher population—the larger the number of individuals, the smaller the effects of genetic drift. These consequences are illustrated in Fig 1 which shows the frequency of one allele in populations of different sizes, $N = 10$ (Fig 1A), $N = 100$ (Fig 1B), $N = 1000$ (Fig 1C) or $N = 10000$ (Fig 1D). Colored lines show trajectories for 8 separate populations evolving over 100 generations. In small populations, random sampling of alleles leads to strong fluctuations in allele frequencies over time. After relatively

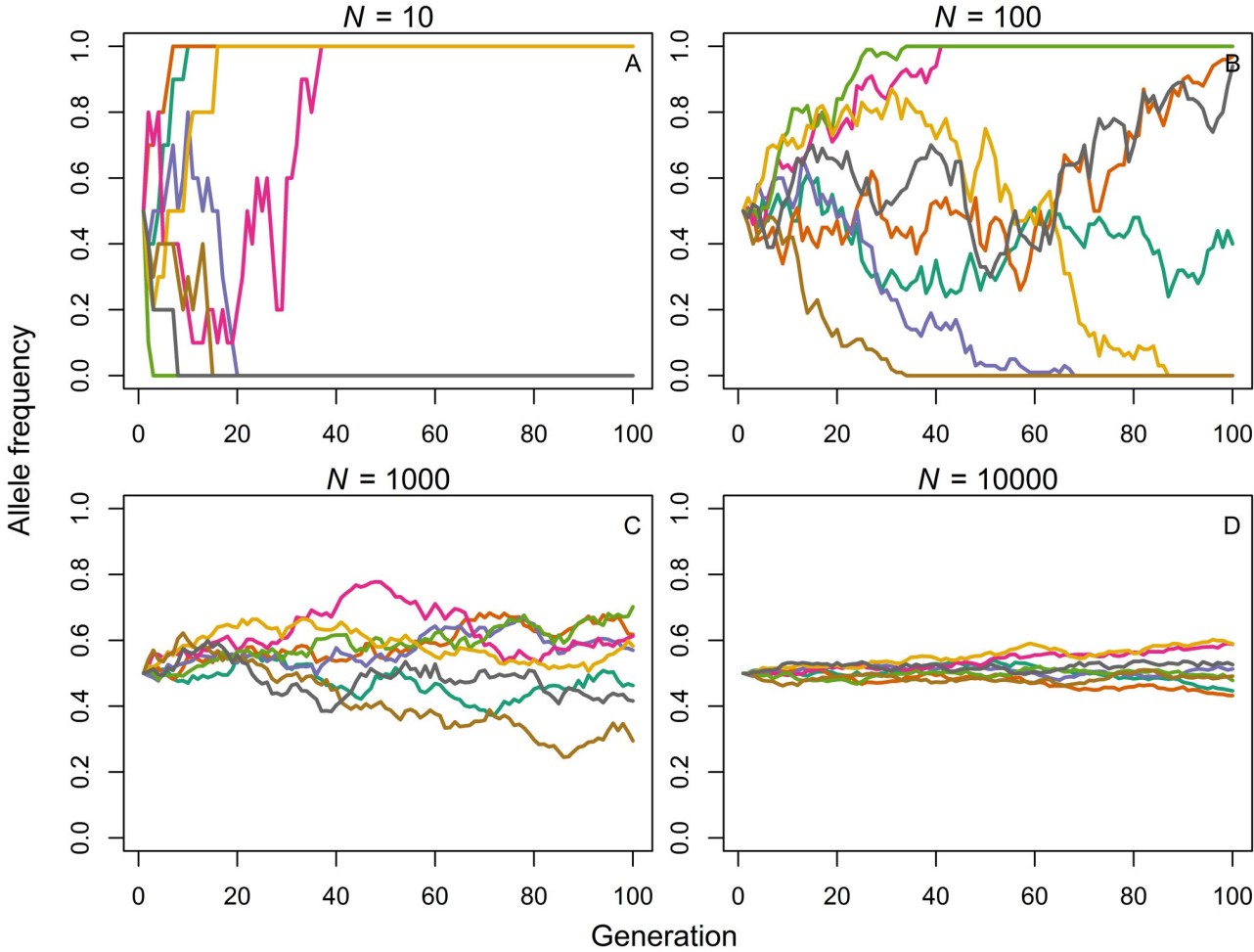

**Fig 1. Drift.** Frequency of a focal allele over 100 generations. Each line shows evolution in a separate population of size $N = 10$ (A), $N = 100$ (B), $N = 1000$ (C) or $N = 10000$ (D). All simulations start at the same initial allele frequency of 0.5.

few generations, populations diverge and the allele either becomes fixed or goes extinct with equal probability given a 0.5 initial frequency [7]. The larger the population, the smaller are temporal fluctuations in allele frequency and the longer it takes for isolated populations to diverge. Given enough time, in the absence of other evolutionary forces, even very large populations will diverge as much as smaller populations. Population size, thus, affects the rate of drift but not its eventual outcome.

## Effective population size

The effective size of a population, $N_e$, is a fundamental concept in population genetics that allows researchers to quantify the effect of drift on evolution [7, 22, 23]. $N_e$ is defined as the size of an idealized Wright-Fisher population that is identical in some key measure of genetic drift to a particular study population. Jointly with the mutation rate, $N_e$ determines the expected number of neutral or weakly selected genetic variants maintained at a locus for a given population and is thus important for correctly calculating the variability in a population. In combination with the strength of selection, $N_e$ also governs how effective selection can be in

spreading favourable mutations and eliminating deleterious ones thus shaping the course of adaptive evolution [23].

Different aspects of the evolution of the Wright-Fisher population have been used to define $N_e$. These most often agree but diverge under some circumstances relevant to cultural systems. Therefore, we consider two commonly used measures, inbreeding effective population size, $N_e^i$, and variance effective population size, $N_e^v$. In S3 Text, we provide derivations for haploid populations for situations when (1) there is variation in offspring numbers and (2) population sizes might differ between parental and offspring generation. The identity-by-descent (or inbreeding) effective population size $N_e^i$ utilizes the fact that, for finite populations, there is a certain probability for two randomly selected individuals to come from the same parent. It is calculated as:

$$N_e^i = \frac{N_{t-1}\bar{k} - 1}{\bar{k} - 1 + \frac{\sigma^2}{\bar{k}}},\tag{1}$$

where $N_{t-1}$ is the census size in the parental generation, $\bar{k}$ is the mean number of offspring and $\sigma^2$ is the variance in offspring number among members of the parental generation.

The variance effective population size $N_e^v$, in contrast, focuses on the amount of random variation in allele frequencies from one generation to the next and can be calculated as:

$$N_e^v = \frac{(N_{t-1} - 1)\bar{k}}{\frac{\sigma^2}{\bar{k}}}.\tag{2}$$

Note that the inbreeding effective number is intimately related to the size of the population in the parental generation, while the variance effective number is related to the size of the population in the offspring generation (see also [12] for equation that uses $N_t$, rather than $N_{t-1}$). To understand why, imagine two parents, one carrying allele $A$, one carrying allele $B$, randomly producing a large number of offspring. In this scenario, two offspring will have a large probability of sharing the same parent (resulting in relatively low inbreeding effective size), but the frequency of both alleles will still be close to 50% in the offspring generation (resulting in relatively high variance effective size). If, on the other hand, a large number of parents produces only a small number of offspring, there will only be a small probability that two offspring share the same parent, while allele frequencies will differ greatly among generations (i.e. high inbreeding effective size and low variance effective size).

The inbreeding effective number is appropriate when researchers are interested in the change in homozygosity due to random drift. The variance effective number, in contrast, is appropriate when researchers are interested in the amount of gene-frequency drift or the increase in variance among subgroups [22, 24]. Despite these differences, for constant population sizes (i.e. $\bar{k} = 1$), both effective numbers agree and equations simplify to:

$$N_e = \frac{N - 1}{\sigma^2}.\tag{3}$$

In the following, we will only differentiate between the two formulations when necessary and otherwise use the simplified version given by Eq (3).

## Methods—Effective population size in cultural evolution

Researchers have identified several factors influencing $N_e$ in genetic evolution (see e.g. [23]). In the case of cultural evolution, the relationship between census population size and the effective size may be considerably more complex. For example, the mode of transmission has been shown to be an important factor in determining effective size in genetic systems. In the case of

culture, there are many more possible modes of transmission [1, 2, 25] each of which may have unique effects. Therefore, in order to correctly use the concept of an effective size in cultural systems, a uniquely cultural theory must be developed.

Traits that evolve culturally do so through processes of innovation and cultural transmission. So, we now consider cultural transmission rather than biological inheritance and allow $\bar{k}$ to represent the average number of naive individuals to which role models transmit their cultural variant. Similarly, $\sigma^2$ represents the variance in this 'cultural influence'. Eq (3) implies that, in general, increasing the variance among individuals in the number of cultural offspring they leave reduces the effective population size, and decreasing that variance increases $N_e$. Thus, it is clear that any process that systematically alters the way in which cultural role models are chosen, through the mode of transmission, demography or social network structure, will change the variance in cultural influence and the effective population size for culturally evolving traits. As noted before, if different traits are passed via different mechanisms within the same population, they might each exhibit different effective population sizes even though the population has the same census size.

## Simulation set-up

As the derivation of analytical results for effective population sizes becomes unfeasible for most of the situations considered in this paper, we develop a simulation framework based on the Wright-Fisher dynamic [26]. We consider a population of census size $N$ where individuals are characterized by the variant of a single cultural trait they have adopted. In each time step, a new generation of individuals is formed and each naive individual adopts its cultural variant, if not specified differently, through unbiased cultural transmission from the previous generation. In more detail, the probability that a naive individual chooses variant $i$ of $M$ cultural variants present in the previous generation is given by:

$$p_i = \frac{n_i}{\sum_{m=1}^{M} n_m}(1 - \mu), \tag{4}$$

where $n_i$ is the frequency of variant $i$ in the appropriate pool of role models. With probability $\mu$ an innovation takes place and a new, not previously seen variant is introduced into the cultural system. To calculate the effective population size in various scenarios, we first let the system evolve through unbiased transmission until it reaches its equilibrium state (see S4 Text for details). We then run 300 generations assuming the transmission dynamics described below and record the "cultural influence" of each individual in a specific generation by determining its number of cultural offspring in the next generation. This provides us with estimates for $\bar{k}$ and $\sigma^2$ (conditioned on the assumed transmission dynamic) needed to calculate effective population sizes according to Eqs (1) and (2). Additionally, to relate effective population sizes to resulting levels of cultural diversity, we record two diversity measures, the Simpson diversity index (SDI) and the number of unique cultural variants. The SDI is calculated as $D = 1 - \sum_{i=1}^{S} (n_i/N)^2$, where $n_i$ is the frequency of individuals carrying variant $i$ in the population and $S$ is the total number of unique cultural variants [27]. This index ranges from 0 to 1, where high scores indicate high cultural diversity (a large number of variants each displayed at a low frequency) and low scores indicate low cultural diversity (few variants, some of which are displayed at relatively high frequencies).

To account for transmission processes different from unbiased transmission, we adapted Eq (4). For *one-to-many transmission*, we assign, in each generation, $R$ individuals at random as role models and record the frequency $n_i$ in Eq (4) only from these $R$ individuals. For *frequency-dependent transmission*, we assume that the probability for adopting variant $i$ of $M$

cultural variants present in the population is given by:

$$p_i = \frac{n_i^\theta}{\sum_{m=1}^{M} n_m^\theta} (1 - \mu), \tag{5}$$

where $n_i$ is the frequency of variant $i$ in the whole population and $\theta$ controls the direction and strength of frequency-dependent bias [28]. When $\theta = 1$, cultural transmission is unbiased; as $\theta$ becomes larger than 1, individuals become increasingly likely to adopt high-frequency variants. When $0 < \theta < 1$, individuals disproportionately adopt low-frequency variants.

To account for *migration* or *cultural exchange*, we use two, initially independent, populations evolving through unbiased transmission. In case of migration, per time step, an average of $mN$ randomly chosen individuals from one population permanently migrate to the other population; they carry their cultural variants with them and consequently serve as potential role models for the next generation. The variable $m$ controls the migration rate. To keep population sizes constant, the same number of individuals immigrates from the other population. In the case of cultural exchange, per time step, an average of $eN$ randomly chosen individuals do not permanently migrate between populations, but are available as additional role models and, thereby, increase the size of the parental generation in both populations. The variable $e$ controls the cultural exchange rate.

To account for *social network structure*, we arrange the $N$ individuals in the population according to different network topologies (random networks, scale-free networks and small-world networks); this restricts the pool of role models for each individual: the probability of choosing cultural variant $i$, $p_i$ as given in Eq (4), is determined only from its direct neighbours in the network. Our aim here is not to replicate realistic social networks but to use prototypical network types to illustrate potential effects of network structure on effective population sizes. Such extreme cases are often useful to identify causal effects and school intuition (see [29, 30] for similar analyses for genetic evolution). All networks considered here are undirected and are generated as follows:

1. Random networks: In random networks, any two individuals have the same probability of being connected. The Erdős-Rényi model generates such a graph by starting with a set of $N$ isolated nodes and creating every possible edge with the same constant probability $p$ [31]. For undirected graphs, there are $\frac{N(N-1)}{2}$ possible ties and $p$ gives the expected proportion of those potential ties that are realized in the network (i.e. the network density).

2. Scale-free networks: A network is said to be scale free if the fraction of nodes with degree $k$ follows a power law $k^{-\alpha}$, where $\alpha > 1$. The Barabási-Albert model is an algorithm that uses a preferential attachment mechanism to generate such networks [32, 33]. Here, it is assumed that new nodes are added to the network two at a time. New nodes are connected to existing node $i$ (out of $J$ total nodes) with a probability $P_i$ that is proportional to the number of links $k_i$ that a node already has: $P_i = \frac{k_i^\pi}{\sum_{j=1}^{J} k_j^\pi}$. That is, well-connected nodes are likely to get even more connected over time. The power of this preferential attachment is controlled by a parameter $\pi$, where $\pi = 1$ produces linear preferential attachment, $0 < \pi < 1$ produces "sub-linear" attachment, and $\pi > 1$ produces "super-linear" attachment.

3. Small-world networks: Small-world networks are graphs with short average path lengths between nodes and a high clustering coefficient. High clustering means that nodes that you are connected to are also likely to be connected to each other (e.g. most of your friends are also friends among themselves). The Watts-Strogatz model creates a small-world network in two basic steps [34]: We start with a lattice of $N$ nodes with each node being connected

to its $K$ closest neighbors on either side. Each edge in the network is then rewired with a certain probability $p_r$ while avoiding duplicates or self-loops. After the first step the graph is a perfect ring lattice. So when $p_r = 0$, no edges are rewired and the model returns a ring lattice. In contrast, when $p_r = 1$, all of the edges are rewired and the ring lattice is transformed into a random graph.

All simulation results shown in the following are based on 1000 independent simulations per parameter combination.

## Results—Determinants of effective population size in cultural evolution

### Process of cultural transmission

We already know from genetic studies that the mode of inheritance can greatly alter effective population size (e.g. [23]). In cultural systems, the ways in which cultural variants can be passed between individuals, from cultural 'parents' to cultural 'offspring', are even more numerous and complex (e.g. [1, 2, 25]). In the following, we analyze how processes of cultural transmission can influence effective population size. In addition to unbiased transmission, we consider two other transmission processes, *one-to-many transmission* and *frequency-dependent transmission*, whose internal dynamics differ in interesting ways. While the number of transmitting individuals per generation is fixed under one-to-many transmission, under frequency-dependent transmission this number emerges from the interplay between the transmission mechanism and the frequency spectrum of the cultural variants.

**One-to-many transmission.**   We define one-to-many transmission as the situation where only a small, pre-defined, number of individuals can transmit their cultural variant to members of the next generation [35, 36]. This transmission process, which might also be called "some-to-many" or "subset-to-many" transmission, may drastically change the variance in cultural influence and, thus, the effective population size depending on the number of role models, $R$. Here, $R$ individuals are eligible to serve as models for the naive generation and pass on their variant via unbiased transmission and $N − R$ individuals are not eligible to pass on their variant. In other words, each generation, we have a transmitting sub-population of size $R$ and a non-transmitting sub-population of size $N − R$. In this case, the variance of cultural influence can be calculated as follows (see S5 Text for the full details):

$$\sigma^2_{\text{OTM}} = \frac{N - 1}{R}. \tag{6}$$

As $R$ increases, i.e. as more role models have the chance to pass on their cultural trait, the variance of cultural influence in the population decreases, and for $R = N$ we recover the expression for the variance of cultural influence in the standard Wright-Fisher model. Thus, the effective population size, $N_e$, for our one-to-many transmission model is simply the size of the transmitting sub-population per generation:

$$N_e = \frac{N - 1}{\sigma^2_{\text{OTM}}} = \frac{N - 1}{\frac{N - 1}{R}} = R. \tag{7}$$

Fig 2A and 2B show the variance in cultural influence, given in Eq (6), and the effective population sizes, given in Eq (7), for different $R$ values. The grey dots represent the mean values of the effective population size generated by the simulation model and, reassuringly, analytical and simulation results match very well. When everyone is a potential role model (i.e. $R$

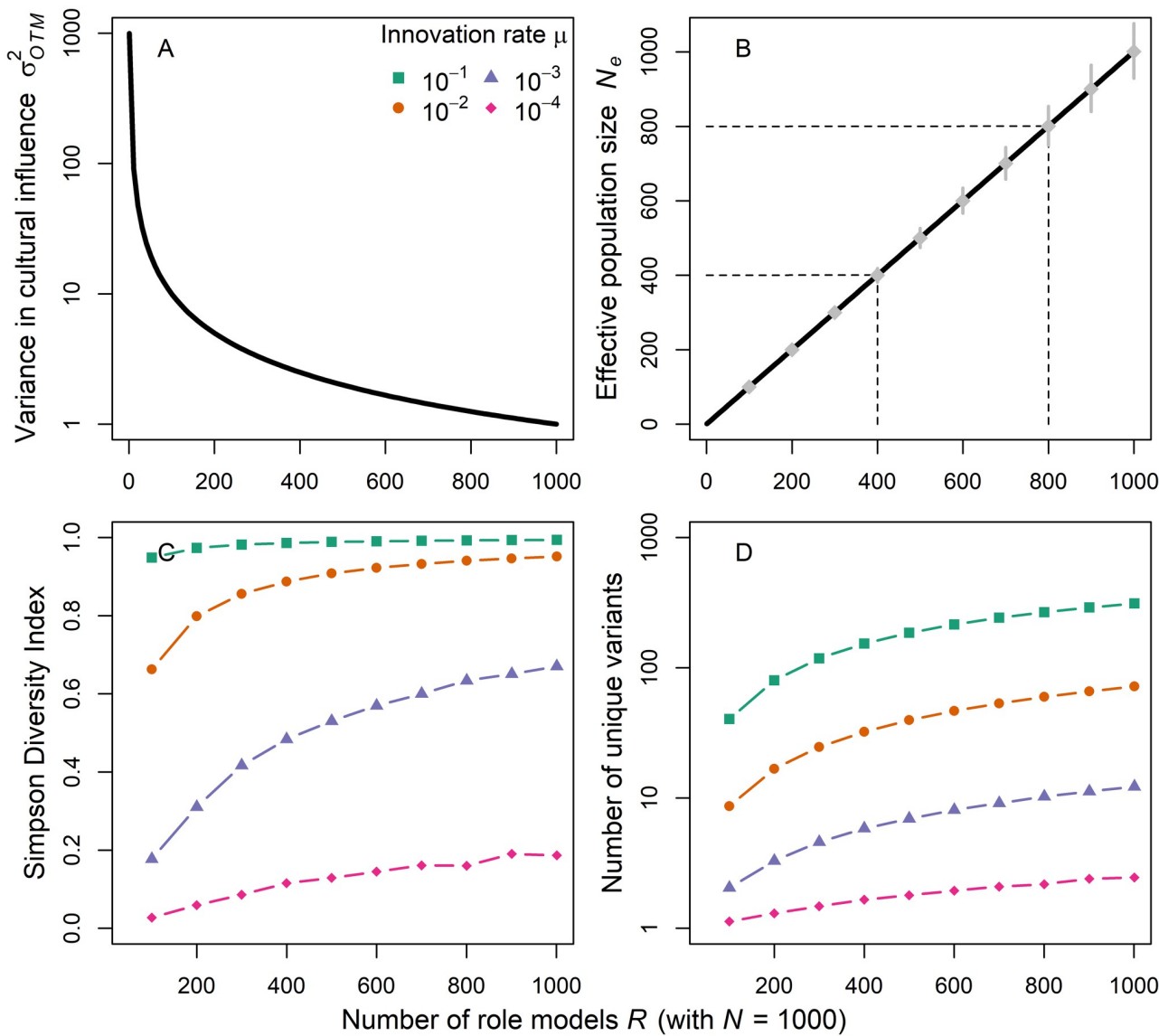

**Fig 2. One-to-many transmission.** Variance in cultural influence ($\sigma^2_{\mathrm{OTM}}$; 2A), effective population size ($N_e$; 2B), mean Simpson diversity (2C) and mean number of unique cultural variants (on log scale; 2D) for different numbers of cultural role models $R$ (with census population size $N = 1000$). Analytical results were confirmed by stochastic simulations. Grey diamonds in plot 2B show means and 90% prediction intervals for 1000 independent simulations (with $\mu = 10^{-4}$).

$= N$), we recover the standard Wright-Fisher model with a variance of cultural influence of approximately 1 and an effective population size of $N_e = N$. Restricting the pool of role models results in an increased variance in cultural influence and decreased effective population size. In the extreme case where the whole population learns from only one individual per generation, the effective population size is 1.

Fig 2C and 2D describe the cultural composition of the population at equilibrium by recording the level of diversity via the Simpson index and number of unique cultural variants in the population. Levels of cultural diversity are jointly determined by the effective population size and innovation rate $\mu$ (census size is held constant at 1000).

**Frequency-dependent transmission.**  We now turn to frequency-dependent cultural transmission where the number of transmitting individuals is not fixed but emerges from the interplay between the transmission process and the frequencies of cultural variants. This form of transmission is well-documented in both human (e.g. [37, 38]) and non-human animals (e.g. [39, 40]). Positive frequency-dependent transmission, or conformity, occurs when the most common variants in a population are disproportionately more likely to be adopted. In contrast, negative frequency-dependent transmission, or anti-conformity, occurs when the rarest variants are disproportionately more likely to be copied.

This dynamic is modelled in our simulation framework through Eq (5). After a burn-in phase under unbiased transmission, i.e. $\theta = 1$, we change the $\theta$ value and record how effective population sizes change over time (see Fig 3). We start by analysing relatively strong

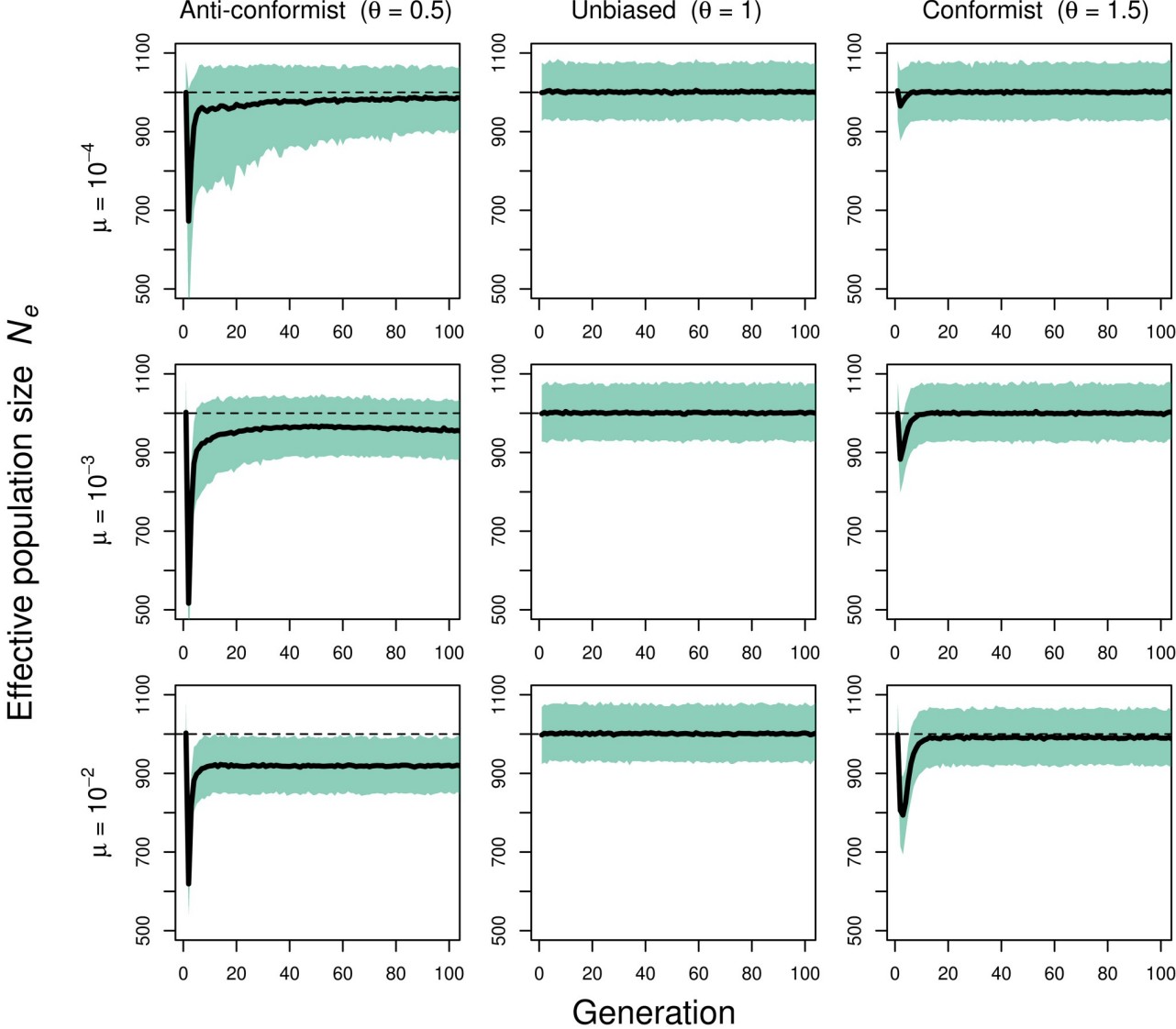

**Fig 3. Frequency-dependent transmission.** Effective population size (including 90% PIs) for anti-conformist ($\theta = 0.5$; left), unbiased ($\theta = 1$; center) and conformist transmission ($\theta = 1.5$; right) and different innovation rates $\mu$. Plots show trajectories for 100 generations after switch in transmission mode (1000 independent simulations; $N = 1000$).

frequency-dependent transmission which results in situations where almost the whole population adopts the same cultural variant (for conformity; $\theta = 1.5$, right column) or all cultural variants have similar frequencies (for anti-conformity; $\theta = 0.5$, left column). Fig 3 shows that the change in transmission process leads to an immediate, and under certain circumstances substantial, decrease in effective population size, followed by a fast recovery. The severity of the trough as well as the new equilibrium after the change is influenced by the innovation rate.

Strong conformity substantially increases the probability that the most common variants are adopted and, therefore, reduces the number of transmitting individuals. This increases the variance in cultural influence and decreases the effective population size. As time progresses, one variant spreads through the population and almost reaches fixation. Because there is no variation in variant frequency for conformity to act on anymore, every individual (at least every individual that does not carry an innovation) is equally likely to pass on their variant to the next generation; this resembles the situation in the standard Wright-Fisher model. Consequently, at equilibrium $N_e \approx N$. We note that in the case of conformity, at equilibrium, higher innovation rates only result in a very slight decrease in effective population size (see Fig 3 bottom row, right) as innovations are quickly driven to extinction.

Strong anti-conformity substantially increases the probability that the rarest variants are copied, again reducing the number of transmitting individuals. As time progresses, variants become equally distributed and, consequently, individuals do not differ greatly in the likelihood of passing their variants to the next generation. But in contrast to the conformist situation, the effective population size at equilibrium is greatly influenced by the innovation rates; per definition, innovations are rare and, thereby, the target of anti-conformity. The higher the innovation rate, the more likely a variant is present in the population at low frequency and, therefore, the higher the differences between individuals in their cultural influence.

Importantly, we note that the dynamics displayed in Fig 3 only occur under relatively strong frequency-dependent transmission. In S1 Fig, we show that weaker forms of frequency-dependent transmission leave effective population sizes largely unchanged as now the change in transmission mode does not generate sufficiently large differences in individuals' likelihood to pass on their cultural variant. In the supplementary material, we also plot time trajectories for the number of unique variants (S2 Fig) and the Simpson diversity index (S3 Fig). Conformist (anti-conformist) transmission decreases (increases) Simpson diversity and number of variants over time, irrespective of the observed time dynamic of the effective number. In fact, our results show that anti-conformist transmission yields *greater* cultural diversity than unbiased transmission (S2 and S3 Figs) even though a trait passed via anti-conformist transmission has a *lower* effective number than one passed via unbiased transmission.

Summarizing, cultural transmission processes different from unbiased transmission do not necessarily lead to a divergence between census and effective population size. This only happens if transmission processes, such as one-to-many and strong frequency-dependent transmission, produce substantial heterogeneity in the probability with which individuals pass on their cultural variant to the next generation, i.e. their cultural influence.

## Population connectedness

In the previous sections, we analyzed the impact of different processes of cultural transmission on the effective size of a cultural population. We now turn to the question of how population properties themselves might influence $N_e$. Empirical tests of demographic hypotheses often consider connectedness among groups, which has been assumed to change the effective size of the populations under consideration [3, 14]. We start by analyzing the effects of population connectedness in the form of migration and cultural exchange.

Fig 4 shows effective population sizes and diversity indices for various degrees of migration on the left and cultural exchange on the right. Irrespective of its rate, migration as we defined it influenced neither inbreeding nor variance effective population size. While introducing new variants into the population, migration in our model does not systematically change the probability individuals get to pass on their cultural traits. As population sizes are constant and individuals still learn from random members of the parental generation, whether they are recent immigrants or not, this scenario corresponds to the standard Wright-Fisher population. Although effective numbers remain unchanged, even small amounts of migration increase both measures of cultural diversity compared to isolated populations. Further raising migration rates leaves diversity largely unchanged indicating that being connected through migration at all has the largest impact on diversity irrespective of the specific rate. Note that if migration increases the census size in a focal population, the effective size tracks this increase (see S4 Fig for a scenario where constant immigration raises the effective size and cultural diversity in a focal population).

Cultural exchange, on the other hand, increases the inbreeding effective population size but not the variance effective size. Why is this? In case of cultural exchange, there are $N(1 + e)$ individuals in the parental generation that pass on their cultural variants to only $N$ individuals in the offspring generation. For the inbreeding effective size, this reduces the probability that two randomly picked individuals copy the same role model in the parental generation as there is now a greater pool of individuals to learn from. In contrast, the variance effective size is more directly related to the number of learners than to the number of role models (see [12], pp. 345–359, for further explanation). In practice, this means that in cases where cultural exchange, or similar processes, are important features of a population's cultural life, the relationship between census population size and effective size will depend on the measure chosen. Finally, increasing cultural exchange causes the greatest increase in Simpson diversity and the number of unique variants when exchange rates are low.

Summarizing, the effects of connectedness on effective population sizes are subtle, probably difficult to detect, and depend on the exact form of connectedness and effective size formulation.

## Social network structure

Real populations are often highly structured in terms of kinship, peer relationships or social class, all influencing who individuals are most likely to interact with and learn from [3, 41, 42]. To reflect individual heterogeneity in potential role models and mimic different ways of information flow, we arrange the $N$ individuals of the population in networks with different properties (see Fig 5, top row, and section on simulation setup for a more detailed description).

Fig 5, second row, shows how the way in which transmissions are structured affects the trait's effective population size. In random networks, irrespective of network density (i.e. the ratio between observed and possible edges), the effective size always equals the census size of the population. As every individual has the same probability $p$ of being connected to any other individual, the number of links are binomially distributed and there is no systematic difference in individuals' probability to pass on their cultural variants. When network connections are random, density also does not affect cultural diversity.

The situation is very different for scale-free networks. Per definition, their power-law degree distribution implies drastically different levels of cultural influence depending on network position. Individuals central to the network will spread their cultural variant to a greater number of individuals while more peripheral individuals will pass on their variant to just a few. This increased variance in cultural influence results in substantially lower effective

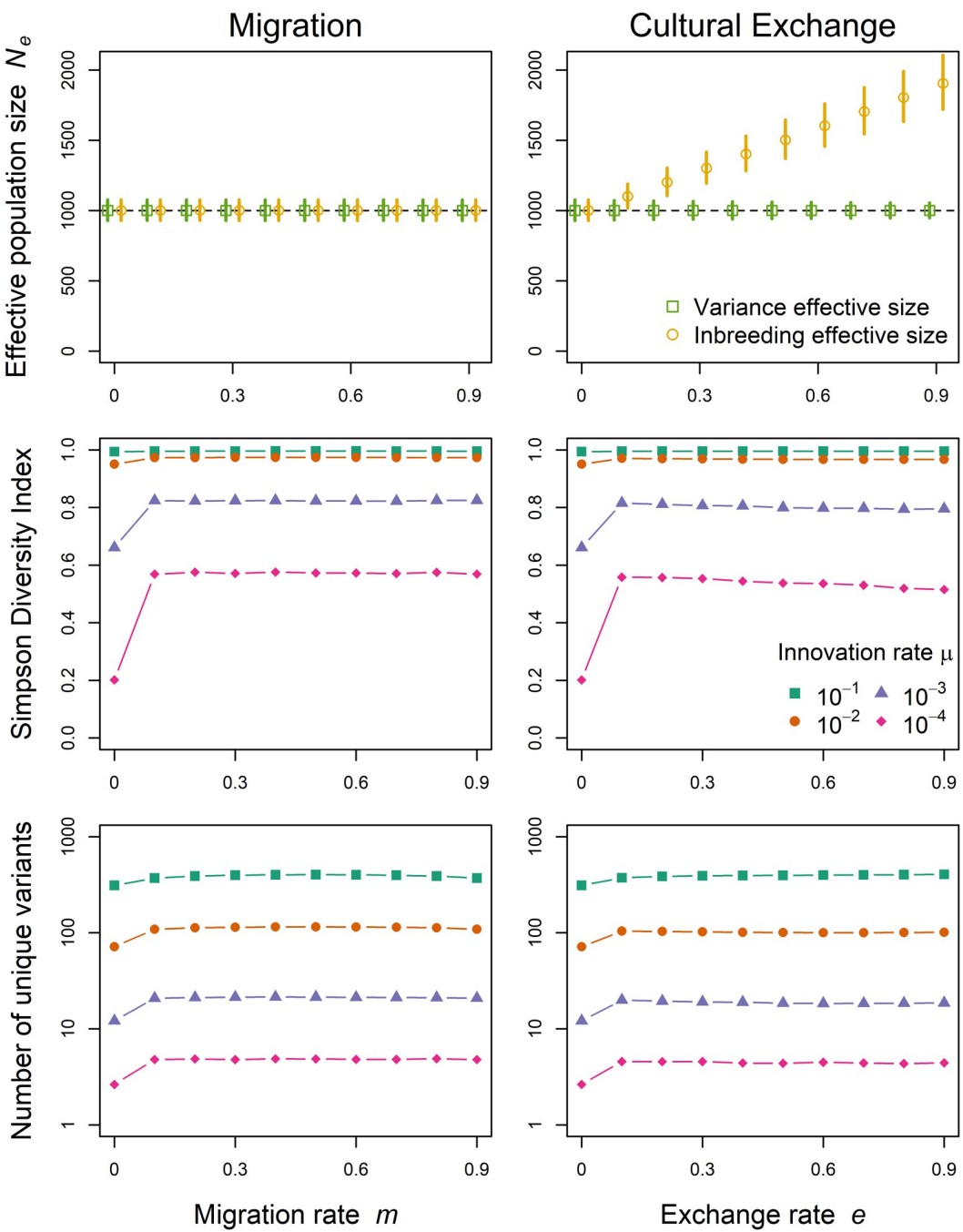

**Fig 4. Population connectedness.** Variance (green squares) and inbreeding (yellow circles) effective population sizes (including 90% PIs; top), Simpson diversity (center) and mean numbers of unique variants (on log scale; bottom) for different migration rates $m$ on the left and cultural exchange rates $e$ on the right. We need to differentiate between effective size formulations because population sizes might differ between parental and offspring generations. Results come from 1000 independent stochastic simulations with census population size $N = 1000$.

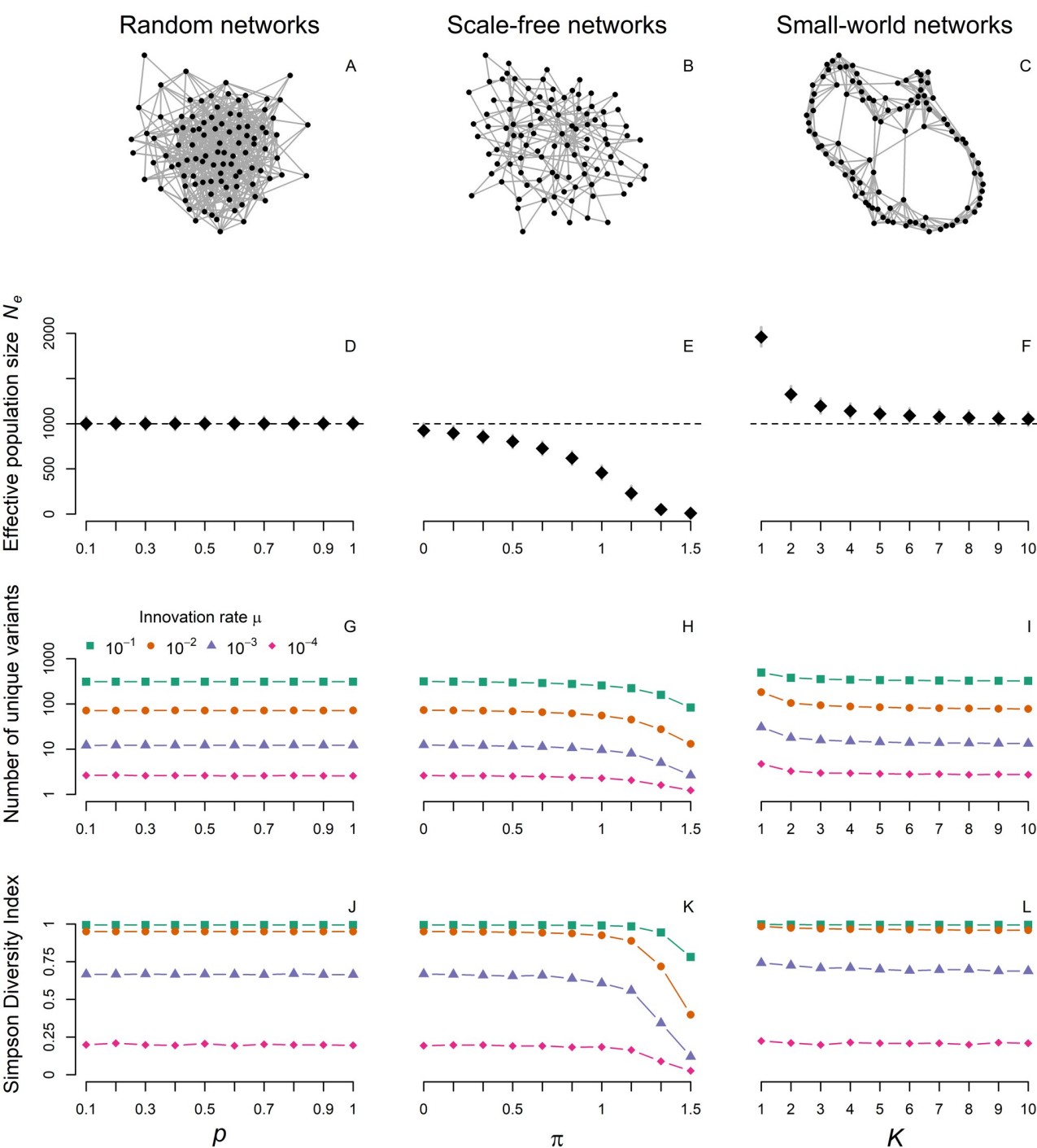

**Fig 5. Social network structure.** Exemplary networks (A-C), effective population sizes (with 90% PIs; D-F), mean numbers of unique cultural variants (on log scale; G-I) and Simpson diversity indices (J-L) for random (Erdős-Rényi), scale-free (Barabási-Albert) and small-world (Watts-Strogatz) networks. Parameter $p$ gives the probability any two nodes are connected in random networks, $\pi$ is the power of preferential attachment creating scale-free networks and $K$ represents the number of initial neighbors on each side in small-world networks (with $p_r = 0.01$). Note that because of structural differences between network types, the ranges of parameter values on the x-axes are not directly comparable. All graphs are created using the *igraph R* package [43]. Results come from 1000 independent simulations with census population size $N = 1000$. On the top, only 100 nodes are drawn for ease of illustration with $p = 0.1$, $\pi = 1$ and $K = 4$.

numbers and also lower levels of cultural diversity. In the extreme case, strong preferential attachment results in a star-shaped network and all individuals will learn from few very central models.

Finally, effective numbers tend to be greater than census numbers for small-world networks. This demonstrates that $N_e$ can also exceed $N$ in cultural systems. As a consequence of strongly local cultural transmission, the variance in cultural influence is reduced compared to random or fully-connected networks. That way, rare cultural variants that would otherwise be quickly lost due to drift, can be shared in local clusters maintaining higher levels of cultural diversity. As either the number of initial neighbors, $K$ (shown here), or the rewiring probability, $p_r$, increases, we approach a fully connected network where everyone can learn from anyone else and $N_e$ approaches census size $N$.

## Discussion

We have systematically examined effective population size, a concept derived from theoretical population genetics, for cultural traits. The effective size allows us to compare populations (or traits), where it would otherwise be difficult to do so. We showed that both modes of cultural transmission and relevant elements of population structure affect the effective population size of a cultural trait. One-to-many and frequency-dependent transmission can substantially lower effective population size with the strongest effects of frequency dependence occurring when the system is out-of-equilibrium. Investigating different forms of connectedness between populations, we found that migration as we define it does not increase $N_e$ and cultural exchange among groups increases inbreeding effective number but not variance effective number. This implies that considerable precision is needed in applying an appropriate cultural effective size formulation. Finally, while random networks with varying densities leave a trait's $N_e$ unchanged, scale-free networks tend to decrease and small-world networks tend to increase $N_e$ compared to the census number.

Population size has been invoked to explain several patterns of cultural change, most notably the emergence and loss of cultural "complexity" observed in the ethnographic and archaeological record (see below for critical discussion of concept of "complexity"). Several theoretical models have been developed to better understand the interplay between learning and demography in generating cultural complexity [16, 17, 44, 45]. Although quite diverse in terms of underlying mechanisms, these models generally agree in predicting more complex cultural repertoires in larger populations. Both real-world ethnographic and archaeological data as well as controlled lab experiments have been used to test the relationship between population size and cultural complexity. Results with both approaches have been mixed with some studies supporting the hypothesis (e.g. [44, 46–49]) but others not (e.g. [50–53]).

These inconsistent findings do not necessarily refute the theoretical models but might instead reflect a poor correspondence between theory and empirical tests (see also section on "Revisiting empirical tests" in [8]). First, "complexity" itself is notoriously difficult to pin down [54] and "cultural complexity" is hence conceptualized very differently across different models and empirical investigations making it difficult to draw inferences. In Henrich's model, for instance, cultural complexity reflects a difficult to learn skill and is represented by the value of a Gumbel-distributed continuous variable $z$ [17], whereas other models describe cultural complexity as cultural repertoire size [45, 55] or the level of innovations in a tree-shaped cultural landscape [56]. Similarly, empirical studies use measures as diverse as toolkit size [46], number of techno-units [46, 50, 57] as well as paper plane flight distances [53]. Instead of the poorly defined notion of cultural complexity, in the present work, we rely on the well-defined and workable concept of cultural diversity.

Here, we have demonstrated that obtaining correct and comparable values for the population size in complex cultural scenarios is not straightforward either. In order for the results from theoretical models to apply correctly to empirical cases, we need to ensure that model parameters are correctly translated into measures from complex real-world scenarios. Our results show that, when there are a few highly influential individuals who—through transmission modes—strongly influence the cultural makeup of the population, the size of the population and the effective size of the cultural trait in question can diverge. Similarly, where populations are organised into social networks in which individuals are heterogeneous with respect to their degree, the ratio between the census size of the population and the effective size of the cultural trait can either increase or decrease depending on network structure. These results also highlight that even relatively small populations might be able to maintain comparatively high levels of cultural diversity if connections are structured in a certain way. Through predominantly local transmission in small-world networks, cultural variants can persist in parts of the network over long periods of time resisting the effects of drift. In this case, larger local clusters or more links between clusters somewhat counter-intuitively reduce the effective population size even though—or rather, because—individuals now share ties with more potential cultural models.

In cultural evolutionary studies, effective size has mostly been invoked as a rhetorical device to explain why the demographic hypothesis fails to predict observed levels of complexity in certain cases (however, see [8] for a first formal approach). Our results suggest that this reasoning is too simplistic. For instance, the effects of interconnectedness depend not only on the exact process of exchange but also on the way effective size is defined. Neither migration nor cultural exchange, as we have modelled them, have consistent effects on the effective population size of the cultural trait. These results do not imply that connectedness between populations is not an important factor for cultural dynamics, but rather that its effect is likely not through increasing the effective size of a population. The finding that small amounts of cultural exchange result in the most diverse populations confirms previous theoretical results suggesting that partial connectivity among populations maximizes cultural accumulation [56, 58].

Overall, these results highlight that census numbers cannot generally be relied on when evaluating hypotheses about the effects of demography on culture. It is the effective size that matters and inferring effective population sizes requires detailed knowledge about underlying cultural and demographic processes on a trait-by-trait basis. To our knowledge, there are no existing methods applicable to estimating effective size in complex cultural systems (for genetic data, see [59] for an approximate Bayesian computation method to infer genome-wide average effective population size). The basic theory of $N_e$ in cultural systems is complicated and in need of considerable development before estimation could become feasible through, for instance, generative inference [60]. Estimation methods will need to take account of several relevant mechanistic deviations from ideal models simultaneously, where, for example, transmission modes and social networks both play important roles.

Our results can also be used as more informal heuristics to decide when census numbers could be used as proxies for the theoretically relevant effective numbers. In societies where cultural influence of a given trait is highly skewed in favor of a small elite, for instance, there is no reason to expect that the overall size of that population should be related to cultural diversity at that trait. In societies where all individuals are more or less equally likely to transmit their ideas and behaviors, census size might be a better approximation of the effective size of a cultural trait. However, note that also most egalitarian societies are characterized by substantial degrees of domain expertise expected to skew cultural influence for certain traits. This again highlights that cultural phenomena are to be studied on a trait-by-trait basis. With respect to transmission biases (or social learning strategies), our results suggest that census numbers

might be used as proxies for the $N_e$ of a trait as long as biases are relatively weak and do not drastically change the relative success of certain cultural variants. Further, if due to recent transformative events, cultural systems are out of equilibrium, researchers should not expect census numbers to conform to the theoretically relevant quantity. Finally, if social network structure imposes limits on cultural transmission, census numbers might still be appropriate to test demographic hypotheses as long as connections are random. Most real-world social ties, however, are unlikely to be random and, in that case, census population size might as well under- or overestimate the effective size of a cultural trait depending on exact network configurations.

In summary, in order to use the concept of effective population size as an explanatory tool in cultural systems, we must first understand how uniquely cultural processes impact its calculation and use this understanding to develop sophisticated estimation methods capable of capturing the complexity of real world cultural dynamics.

## Supporting information

**S1 Text. Shows how to calculate effective population size and expected number of cultural traits for the simple example used in section "What is effective population size and why do we need it?".**
(PDF)

**S2 Text. Explains why both mean and variance in offspring number are approximately 1 for in the haploid Wright-Fisher population.**
(PDF)

**S3 Text. Shows the derivation of both inbreeding and variance effective population sizes as appropriate for cultural evolution.**
(PDF)

**S4 Text. Describes the detailed procedure of the burn-in period for simulation analyses.**
(PDF)

**S5 Text. Derives the appropriate formula for calculating the pooled variance for transmitting and non-transmitting sub-populations that is used to calculate effective population size for one-to-many transmission.**
(PDF)

**S1 Fig. Frequency-dependent transmission for wider ranges of conformity exponent $\theta$.** Effective population size (including 90% PIs) for different levels of anti-conformist ($\theta < 1$), unbiased ($\theta = 1$) and conformist transmission ($\theta > 1$) and different innovation rates $\mu$. Plots show trajectories for 100 generations after switch in transmission mode (1000 independent simulations; $N = 1000$).
(TIF)

**S2 Fig. Cultural diversity under frequency-dependent transmission.** Number of unique variants (including 90% PIs) for anti-conformist ($\theta = 0.5$; left), unbiased ($\theta = 1$; center) and conformist transmission ($\theta = 1.5$; right) and different innovation rates $\mu$. Plots show trajectories for 300 generations after switch in transmission mode (1000 independent simulations; $N = 1000$).
(TIF)

**S3 Fig. Cultural diversity under frequency-dependent transmission.** Simpson Diversity Index (including 90% PIs) for anti-conformist ($\theta = 0.5$; left), unbiased ($\theta = 1$; center) and

conformist transmission ($\theta = 1.5$; right) and different innovation rates $\mu$. Plots show trajectories for 300 generations after switch in transmission mode (1000 independent simulations; $N = 1000$).
(TIF)

**S4 Fig. Alternative migration mechanism that gradually increases the census size $N$ in a focal population.** We start by letting a large source population with $N = 10000$ and a small focal population with $N = 100$ evolve separately until they reach equilibrium; each generation, we then let a fixed number of individuals migrate from the source population to the focal population and record effective numbers and diversity indices in the focal population. Effective population size (left), number of unique cultural variants (on log scale; center) and Simpson Diversity (right) for different innovation rates $\mu$. Plots show trajectories for 150 generations after immigration starts (1000 independent simulations with 50 immigrants per generation).
(TIF)

## Acknowledgments

We thank members of the Department for Human Behavior, Ecology and Culture at the Max Planck Institute for Evolutionary Anthropology in Leipzig for constructive discussions and criticisms which helped improving this paper.

## Author Contributions

**Conceptualization:** Dominik Deffner, Anne Kandler, Laurel Fogarty.

**Formal analysis:** Dominik Deffner, Laurel Fogarty.

**Investigation:** Dominik Deffner, Anne Kandler, Laurel Fogarty.

**Methodology:** Dominik Deffner, Anne Kandler, Laurel Fogarty.

**Software:** Dominik Deffner.

**Supervision:** Anne Kandler, Laurel Fogarty.

**Visualization:** Dominik Deffner.

**Writing – original draft:** Dominik Deffner.

**Writing – review & editing:** Anne Kandler, Laurel Fogarty.

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
