## [Decision Letter · Decision Letter 0]

2 Dec 2021

Dear Dr. Deffner,

Thank you very much for submitting your manuscript "Effective population size for culturally evolving traits" for consideration at PLOS Computational Biology. As with all papers reviewed by the journal, your manuscript was reviewed by members of the editorial board and by several independent reviewers. The reviewers appreciated the attention to an important topic. Based on the reviews, we are likely to accept this manuscript for publication, providing that you modify the manuscript according to the review recommendations.

As you see, Reviewer #1 recommends Accept, while Reviewer #2 recommends Minor Revision. Reviewer #2 has suggested many comments/suggestions/questions and I believe they can help improving the manuscript. Some comments are straightforward, but the others are more conceptual. Thus, I would like to invite re-submission of the revised ms and give it another round of review, at least by Reviewer #2.

Sincerely,

Joe Yuichiro Wakano

Guest Editor

PLOS Computational Biology

Ville Mustonen

Deputy Editor

PLOS Computational Biology

[LINK]

As you see, Reviewer #1 recommends Accept, while Reviewer #2 recommends Minor Revision. Reviewer #2 has suggested many comments/suggestions/questions and I believe they can help improving the manuscript. Some comments are straightforward, but the others are more conceptual. Thus, I would like to invite re-submission of the revised ms and give it another round of review, at least by Reviewer #2.

Reviewer's Responses to Questions

**Comments to the Authors:**

Reviewer #1: This is a very clearly written paper that makes a significant contribution to the ongoing debate about the extent to which population size influences cultural diversity. The authors show that a range of different factors affect the relationship and impact the the relationship between census population size and culturally-effective size, which can be both smaller and larger than the census size. To characterise the relevant factors in empirical cases will be difficult but this paper takes the theory forward significantly.

Reviewer #2: Please see attached pdf file for comments to authors.

**Have the authors made all data and (if applicable) computational code underlying the findings in their manuscript fully available?**

Reviewer #1: Yes

Reviewer #2: Yes

PLOS authors have the option to publish the peer review history of their article (what does this mean?). If published, this will include your full peer review and any attached files.

Reviewer #1: No

Reviewer #2: **Yes: **L. S. Premo

Figure Files:

Data Requirements:

Reproducibility:

References:

---

## [Decision Letter · Decision Letter 1]

10 Feb 2022

Dear Dr. Deffner,

Thank you very much for submitting your manuscript "Effective population size for culturally evolving traits" for consideration at PLOS Computational Biology. As with all papers reviewed by the journal, your manuscript was reviewed by members of the editorial board and by several independent reviewers. The reviewers appreciated the attention to an important topic. Based on the reviews, we are likely to accept this manuscript for publication, providing that you modify the manuscript according to the review recommendations.

As you will see, the reviewer (Luke Premo, he waived his anonimity) found a big improvement in the revised manuscript. In addition to many editorial changes, he suggests some points to improve the manuscript even better. I will send you back the manuscript so that you can make changes. Ater receiving your final version, I will be glad to accept the paper.

Sincerely,

Joe Yuichiro Wakano

Guest Editor

PLOS Computational Biology

Ville Mustonen

Deputy Editor

PLOS Computational Biology

[LINK]

As you will see, the reviewer (Luke Premo, he waived his anonimity) found a big improvement in the revised manuscript. In addition to many editorial changes, he suggests some points to improve the manuscript even better. I will send you back the manuscript so that you can make changes. Ater receiving your final version, I will be glad to accept the paper.

Reviewer's Responses to Questions

**Comments to the Authors:**

Reviewer #2: The revised paper is a big improvement on what was already a very interesting study. The authors did a nice job addressing the concerns I had shared in my initial review. I have included one more round of suggestions and comments that the authors might like to address in their final version. Most of the suggestions are merely editorial, but a few are substantive. My kudos to the authors! I hope some anthropologists find this article.

Line by line suggestions for the final version; these are most cosmetic/grammatical to help clarify the writing:

line 34: “…unless the ways in which the trait of interest is passed in these real populations are otherwise identical in evolutionarily important ways. Put differently, two populations with the same census size…”

line 52: “…the expected number of unique variants of a cultural trait passed via ______ transmission with a copy error rate of ______ is 223.”

line 71: The human population is now close to 8 billion (shockingly), not 7 billion. The 2007 publication cited to support the 7 billion number is now 15 years old and not very close to “the time of publication.” I’d recommend updating this number and dropping the 2007 citation.

line 200: “…we now consider cultural transmission rather than biological inheritance and…”

line 201: “…individuals to which role models transmit their cultural variant.”

line 298: “…variants can be passed between individuals,…”

line 305: “…transmission, under frequency-dependent transmission this number emerges from the interplay between the transmission mechanism and the frequency spectrum of the cultural variants.”

Figure 2 is a great figure. That should be in a textbook.

line 381: “…trajectories for the number of unique variants (Fig. S2) and the Simpson diversity index (Fig. S3).”

line 384: Cool new result. I think this is a great illustration of how cultural transmission can create a big disconnect between Ne and cultural diversity in ways that don’t really happen in genes, and this fits with the authors’ larger goal in this paper. Adding one more sentence here tying figures S1 and S2 together would make an even greater impression and really drive home this point. If it were me, I would heavily emphasize this finding. I would recommend adding something like this to make this important point clear: “In fact, our results show that anti-conformist biased transmission yields *greater* cultural diversity than unbiased transmission (Figs. S2 and S3) even though a trait passed via anti-conformist biased transmission has a *lower* effective number than one passed via unbiased transmission (Fig. S1), holding all else constant.”

Boom!

line 439: “When network connections are random, density does not affect cultural diversity.”

line 444: “…a greater number of individuals…”

line 460: “…populations (or traits) where it would otherwise…”

line 469: “…leave a trait’s Ne unchanged, scale-free…”

line 471: “…several patterns of cultural change,”

line 489-490: Well yes, z is a continuous variable, but Henrich clearly goes a bit further than that. Henrich is clear that z represents a skill that is difficult to copy even when learning from the previous generation’s best practitioner. This idea that z represents a difficult to learn skill motivates his use of the Gumbel distribution. So, in Henrich’s model z is a bit more than “just” a continuous variable, which could be represented by a normal distribution or a uniform distribution or some other distribution. The notion of skill is central to what z means in his model and how noise in transmission is represented with the Gumbel distribution—calling z merely a “continuous variable” misses this point. Others working around the same time seem to have taken the notion that greater skill is the same thing as greater complexity (I don’t think that is necessarily true, and the proposed connection between skill and the size of one’s toolkit was always a mystery to me—toolkit size is a function of many things, "skill" probably among the least important) and it rather quickly spun out of control with a series of empirical "tests" that made use of others’ data and others’ units of analysis that were no longer really addressing anything of substance, at least regarding the effects of “demography” on “complexity”. It is no wonder many of these studies yield confusing and contradictory results—looking back, it seems to me they were interpreting noise rather than signal. In sum, I think the authors can describe z better than just a continuous variable—it was a very particular kind of continuous variable—based on the notion of a transmittable skill—and that helps explain some of the work that followed (and in my opinion, why it went down the wrong track).

line 503: “…size of the population and the effective size of the cultural trait in question can diverge.”

line 505: “…between the census of the population and the effective size of the cultural trait can either…”

line 511: “…reduce the effective population size of the cultural trait even though—or rather, because—individuals now share ties with more potential models.”

line 519: “…on the effective population size of the cultural trait.”

line 520: “…dynamics, but rather that its…”

line 526: “It is the effective size of the trait that matters…”

line 528: “…demographic processes on a trait-by-trait basis.”

line 538: “…cultural influence of a given trait is highly skewed…”

line 539: “…related to cultural diversity at that trait.”

line 541: “…be a better approximation of the effective size of a cultural trait.”

line 546: “…proxies for the Ne of a cultural trait as long as…”

line 548: “…cultural systems are out of…”

line 551: “…connections are random.”

As usual, I waive my anonymity.

Sincerely,

Luke Premo

**Have the authors made all data and (if applicable) computational code underlying the findings in their manuscript fully available?**

Reviewer #2: Yes

PLOS authors have the option to publish the peer review history of their article (what does this mean?). If published, this will include your full peer review and any attached files.

Reviewer #2: **Yes: **L. S. Premo

Figure Files:

Data Requirements:

Reproducibility:

References:

---

## [Editor Report · Decision Letter 2]

21 Feb 2022

Dear Dr. Deffner,

We are pleased to inform you that your manuscript 'Effective population size for culturally evolving traits' has been provisionally accepted for publication in PLOS Computational Biology.

Best regards,

Joe Yuichiro Wakano

Guest Editor

PLOS Computational Biology

Ville Mustonen

Deputy Editor

PLOS Computational Biology

---

## [Editor Report · Acceptance letter]

6 Apr 2022

PCOMPBIOL-D-21-01634R2 

Effective population size for culturally evolving traits

Dear Dr Deffner,

I am pleased to inform you that your manuscript has been formally accepted for publication in PLOS Computational Biology. Your manuscript is now with our production department and you will be notified of the publication date in due course.

With kind regards,

Agnes Pap
